# Accuracy of GynTect^®^ Methylation Markers to Detect Recurrent Disease in Patients Treated for CIN3: A Proof-of-Concept Case-Control Study

**DOI:** 10.3390/cancers16173022

**Published:** 2024-08-30

**Authors:** Heike Hoyer, Cornelia Scheungraber, Grit Mehlhorn, Ingke Hagemann, Sarah Scherbring, Linn Wölber, Annett Petzold, Kristina Wunsch, Martina Schmitz, Monika Hampl, Gerd Böhmer, Peter Hillemanns, Ingo B. Runnebaum, Matthias Dürst

**Affiliations:** 1Institut für Medizinische Statistik, Informatik, Datenwissenschaften (IMSID), Universitätsklinikum Jena, 07743 Jena, Germany; heike.hoyer@med.uni-jena.de; 2Frauenarztpraxis, Westbahnhofstraße 2, 07745 Jena, Germany; scheungraber@outlook.com; 3Frauenarztpraxis, Neustädter Kirchenplatz 1A, 91054 Erlangen, Germany; info@dysplasie-erlangen.de; 4abts+partner Partnerschaftsgesellschaft, Prüner Gang 7, 24103 Kiel, Germany; i.hagemann@abts-partner.de; 5Fachärzte für Frauenheilkunde und Geburtshilfe, Karrenführerstraße 1-3, 38100 Braunschweig, Germany; info@gyn-braunschweig.de; 6Klinik für Gynäkologie, Universitätsklinikum Hamburg-Eppendorf, 20246 Hamburg, Germany; lwoelber@uke.de; 7Klinik und Poliklinik für Frauenheilkunde und Fortpflanzungsmedizin, Universitätsklinikum Jena, 07747 Jena, Germany; annett.petzold@med.uni-jena.de (A.P.); ingo.runnebaum@med.uni-jena.de (I.B.R.); 8Oncgnostics GmbH, Löbstedter Str. 41, 07749 Jena, Germany; kristina.wunsch@oncgnostics.com (K.W.); martina.schmitz@oncgnostics.com (M.S.); 9Universitätsfrauenklinik Düsseldorf, Moorenstrasse 5, 40225 Düsseldorf, Germany; monika.hampl@hohenlind.de; 10Institut für Zytologie und Dysplasie (IZD), Theaterstr. 14, 30159 Hannover, Germany; boehmer@izd-hannover.de; 11Klinik für Frauenheilkunde und Geburtshilfe, Medizinische Hochschule Hannover (MHH), 30625 Hannover, Germany; hillemanns.peter@mh-hannover.de

**Keywords:** DNA methylation, hrHPV, cervical intraepithelial neoplasia (CIN), post-treatment surveillance, recurrent CIN

## Abstract

**Simple Summary:**

Women treated for cervical precancers (CIN2/3) have an increased risk for recurrent lesions and therefore require post-treatment monitoring. Standard follow-up care based on hrHPV-DNA/cytology co-testing has high sensitivity but limited specificity. Host cell methylation correlates highly with CIN2/3 and may thus be a more suitable marker than hrHPV, particularly since HPV infection is not confined to women with the disease. Here, we show that in combination with cytology, the methylation markers comprising GynTect^®^ are as sensitive as hrHPV for the detection of recurrent CIN2/3 lesions but are more specific. A reduction in the rate of false positive results would not only be more cost-effective but may also reduce the number of women requiring re-treatment. These findings must be verified in a sufficiently large validation study.

**Abstract:**

Post-treatment follow-up in women with CIN3 is mandatory due to relapse in up to 15% of patients within 2 years. Standard follow-up care based on hrHPV-DNA/cytology co-testing has high sensitivity but limited specificity. The aim of our proof-of-concept case-control study was to evaluate the performance of the methylation test GynTect^®^ for the detection of recurrent CIN2/3 during follow-up. Residual clinical material from a recent, prospective, multicenter, observational study was available for further analysis. We studied a sample of 17 cases with recurrent CIN2/3 diagnosed within 24 months of follow-up and 31 controls without recurrence. DNA from cervical scrapes at baseline (immediately before CIN3 surgery) and up to three follow-up visits were analyzed for hrHPV and GynTect^®^ methylation status. Cytology data were available from the previous study. Overall, 12 cases and 21 controls were GynTect-positive at baseline. In these subgroups, single test sensitivity at first follow-up was 67% (95% CI 39–87%) for GynTect^®^ compared to 83% (95% CI 55–96%) for hrHPV (*p* = 0.50). Single test specificity was significantly higher for GynTect^®^ (90%, 95% CI 71–98% vs. 62%, 95% CI 40–80%) (*p* = 0.03). In a co-testing setting, both hrHPV/cytology and GynTect^®^/cytology detected all recurrences. Specificity for GynTect^®^/cytology was higher than for hrHPV/cytology, but this difference was not statistically significant. In conclusion, for initially GynTect-positive patients, both hrHPV and GynTect^®^ tests detected recurrent disease with similar sensitivity, but the GynTect^®^ assay has a higher specificity. Incident hrHPV infection and/or persisting multifocal hrHPV infections without clinical disease are most likely responsible for the poorer specificity of the hrHPV test. A future prospective validation study will have to show whether GynTect^®^/cytology co-testing can outperform hrHPV/cytology co-testing in post-treatment surveillance.

## 1. Introduction

Cervical cancer is essentially a virus-induced disease. Already at the turn of the 20th century, the first large prospective studies revealed the great promise human papillomavirus (HPV) diagnostics would have for the efficient detection of high-grade cervical epithelial lesions (CIN2/3) and cervical carcinoma [1,2,3]. Twenty years later, most countries in Europe have implemented high-risk (hr)HPV diagnostics as a primary cervical cancer screening test [4]. Since an HPV-positive test result does not allow us to distinguish between a mere viral infection and clinically relevant disease, HPV-positive cases are usually triaged by cytology. Because of the lack of reliable prognostic markers, CIN2/3 is still an indication for surgical treatment. Conization by loop electrosurgical excision procedure (LEEP) is the most widely used technique [5]. The main advantage of LEEP, as compared to laser ablation, is that it allows a pathologic examination of the removed lesion in its entirety. However, despite treatment, up to 15% of patients develop recurrent high-grade disease within 2 years of follow-up [6,7,8,9]. Recurrent disease either reflects residual disease due to incomplete removal of the initial lesion or incident (new) disease. Actually, most of the recurrent disease diagnosed within 2 years is residual disease [10]. Risk factors that correlate with recurrent disease include hrHPV genotype, viral persistence, positive resection margins, endocervical crypt involvement, lesion size, and lesion severity [11,12,13,14,15]. In light of the substantial risk of recurrent disease, effective post-treatment monitoring, also referred to as a test of cure (ToC), is crucial. In the case of post-treatment care, the best sensitivity of 95% (95%CI 91–98%) was achieved by hrHPV and cytology co-testing, albeit at the expense of specificity of only 67% (95% CI 60–74%) [16]. An algorithm comprising co-testing after 6, 12, and 24 months is now the standard of care in many countries, including Germany [17,18,19]. The relatively high false positive rate of course gives rise to unnecessary patient anxiety and the risk of overtreatment. By testing for hrHPV-RNA rather than hrHPV-DNA, specificity could be improved while maintaining high sensitivity [20,21]. Besides hrHPV/cytology co-testing, the performance of several other markers for the monitoring of women treated for high-grade CIN is being explored. The clinical usefulness of diagnostic markers such as methylation signatures relies on the fact that methylation is characteristic of cancer cells and their precursors, and increased levels correlate with disease development [22,23]. In a recent publication by Dick and colleagues, the methylation markers *ASCL1*/*LHX8* and *FAM19A4*/*miR124-2* reliably detected recurrent CIN2/3 lesions [24]. Moreover, a significant increase in specificity for recurrent disease was observed for p16/Ki67/hrHPV co-testing in comparison to hrHPV/cytology co-testing [25,26]. A further improvement in terms of specificity was expected by the use of individualized molecular markers: A characteristic step in cervical carcinogenesis is that viral DNA frequently integrates into the host genome, thereby promoting clonal cell expansion [27,28,29,30,31,32]. Because the viral–host junction sites are unique for each patient, their DNA sequence information allows for the design of an individualized PCR for the detection of recurrent disease [33,34]. This strategy was evaluated in a prospective, multicenter, observational study to test the hypothesis that an individualized viral–host junction PCR combined with cytology has a lower false positive rate for the prediction of recurrent CIN2/3 compared to standard hrHPV/cytology co-testing. Although absolutely specific on its own, the viral–host junction PCR could not detect all recurrent CIN2/3 in a co-testing setting [9]. This finding may be explained by multifocal lesions, intratumoral heterogeneity with respect to HPV integration, and/or incident CIN. Residual clinical material from the above well-characterized cohort was available for our current retrospective proof-of-concept case-control study to evaluate the accuracy of GynTect^®^ methylation markers versus hrHPV testing for the detection of recurrent disease in patients treated for CIN3. The GynTect^®^ assay is a molecular diagnostic test detecting six DNA methylation markers on human genomes. DNA methylation of these markers correlates with high-grade cervical lesions and cervical cancer [23]. Moreover, in a recent prospective longitudinal study, an initially negative GynTect^®^ result was shown to be highly protective for incident CIN2+ over a period of 3 years [35]. Additionally, we explored the accuracy of the clinically relevant co-testing of the GynTect^®^ assay with cytology.

## 2. Materials and Methods

### 2.1. Initial Cohort, Case, and Control Group Selection and Samples

This case-control study is based on our previous prospective, multicenter, observational cohort study in which the hypothesis was tested that an individualized viral–cellular junction test combined with cytology has a lower false positive rate for the prediction of recurrence compared to standard hrHPV/cytology co-testing [9]. Four hundred forty-five patients with histologically confirmed HPV16 or 18 positive CIN3 were included in the initial cohort provided they underwent surgery for the first time (re-conization was an exclusion criterion), were aged at least 18 years, and had given informed consent. Of 408 patients with valid follow-up data, 20 presented CIN2/3 recurrences within 24 months of follow-up. All cases and about 10% of patients without recurrence were considered for the proof-of-concept study if they had given informed consent for the use of residual material in further studies and revealed valid GynTect^®^ test results at baseline. Thus, a total of 17 cases with recurrent CIN2/3 (7 CIN2, 10 CIN3) and 31 controls, altogether representing eight study sites, were available for this study. The median age of the cases and controls was 31 (range 27 to 49) and 31 (range 25 to 53), respectively. The median time for the diagnosis of recurrent disease was 6 months (range 3 to 24 months). Further characteristics of the patients are shown in the Appendix A. Of note is that the frequency of positive resection margins is higher among cases (29%) versus controls (12%). However, this difference is not statistically significant (*p* = 0.23; Fisher’s exact test). For all 48 patients, DNA from cervical scrapes at baseline (immediately before CIN3 surgery) and at least the first of at most three follow-up visits (6, 12, and 24 months post-surgery) were available (median follow-up 2 years). This DNA was used for hrHPV detection and GynTect^®^ methylation analyses. Cytology data from conventional or thin-layer liquid-based cytology (according to the local standard) was available from the previous study and was classified according to Munich 3 nomenclature. The results were grouped as follows: 0 = insufficient material, negative = Pap I, II-a, II-*p*, II-g, and II-e, or positive = ≥III. In terms of the Bethesda system, Pap I and II-a correspond to NILM, II-*p* to ASC-US, II-g to AGC endocervical NOS, II-e to endometrial cells, and ≥III-*p*, III-g, III-e, III-x to ASC-H, AGC, or higher.

### 2.2. hrHPV DNA Detection

For all patients, the HPV status was determined using the GP5+/6+ PCR-EIA assay [36,37]. This assay allows for the detection of all 13 hrHPV types (HPV16, 18, 31, 33, 35, 39, 45, 51, 52, 56, 58, and 59 and 68) either as group or in a more stratified approach as individual genotypes. Moreover, the assay also allows for the detection of IARC group2B comprising HPV26, 53, 66, 67, 70, 73, and 82, which are considered to be possibly carcinogenic to humans [38]. The accuracy analysis of this study is based on the detection of hrHPV as a group. HPV genotyping was omitted because the determination of individual HPV genotypes is not part of most post-treatment algorithms. HPV genotyping was only performed for patients with recurrence in a post-hoc setting.

### 2.3. Methylation Analysis

All methylation analyses (GynTect^®^ assay) were conducted in the laboratory of oncgnostics GmbH, Jena. All samples were blinded for hrHPV status, cytology, and histology outcomes. The GynTect^®^ assay was performed according to the instructions for use provided by the supplier (oncgnostics GmbH, Jena, Germany). Briefly, 500 ng DNA of each sample was treated with bisulfite using the EpiTect^®^ Fast DNA Bisulfite Kit (Qiagen, Hilden, Germany). After elution of the bisulfite-converted DNA with 20 μL water, 70 μL water was added, and 10 μL was applied to each single reaction in the GynTect^®^ real-time methylation-specific PCR (qMSP) assay. The qMSPs were run on a Cobas Z480 Realtime PCR system (Roche Diagnostics, Alameda, USA). For each marker, the Ct-value was determined, and a delta Ct was calculated between the Ct-value of the quality control marker IDS (prerequisite Ct value IDS ≤ 32.00) and the Ct-value for each marker. For a positive test result, the delta Ct has to be ≤8 for ASTN1, ≤9 for DLX1, ITGA4, RXFP3, SOX17, and ≤10 for ZNF671. If positive, each marker scores as follows: DLX1 score 1; ASTN1, ITGA4, RXFP3, SOX17 each score 2; and ZNF671 score 6. GynTect^®^ is considered to be positive if the total score is equal to or higher than 6 [23,39].

### 2.4. Statistical Analyses

This case-control study was based on our previous prospective, multicenter, observational predictive accuracy study. Due to the proof-of-concept purpose, we refrained from a formal sample size calculation. All eligible cases were included in this study. About 10% of the original cohort without recurrence were used as controls. The ratio of controls to cases was thus about 2 to 1. In practice, every 10th patient without recurrence was chosen in descending order from the list of the original cohort. We estimated sensitivity within cases and specificity within controls using standard definitions in the sub-sample of patients with positive GynTect^®^ results immediately before CIN3 surgery (baseline). Cytology, hrHPV, GynTect^®^ assay, and the combinations of cytology with hrHPV and cytology with GynTect^®^ were considered as index tests. We aimed to predict recurrence from the first follow-up visit six months after surgery (single index test performance) and from all follow-up visits before or at detection of recurrence (program performance). Results of combined tests (co-tests or follow-up tests) were defined as positive if at least one component revealed a positive result (believe-the-positive rule). Wilson 95% two-sided confidence intervals were calculated for the binomial proportions [40]. For the paired comparison we applied the exact McNemar test with a two-sided significance level of 5%. If both tests, which are compared, perform best (100%), the p-value could not be calculated due to zero cells. SAS 9.4 was used for statistical analyses.

## 3. Results

Overall, 15 of 48 patients were GynTect-negative at baseline. For the sub-sample of 33 GynTect-positive patients (12 of 17 cases and 21 of 31 controls), the frequency distribution of cytology, GynTect^®^, and hrHPV test results at the first follow-up visit and the overall results of all follow-up visits is shown in Table 1. The sensitivity and specificity of these three index tests along with 95% confidence intervals are displayed in Table 2.

There was no significant difference in sensitivity when comparing GynTect^®^ and hrHPV results, irrespective of the number of follow-up visits (Table 2). However, the specificity was significantly higher for GynTect^®^ (90%, 95%CI 71–98%) when compared to hrHPV (62% (95%CI 40–80%)) (McNemar *p* = 0.03) at the first follow-up visit and was just below significance for all follow-up visits (Table 2).

The performance of co-testing is shown in Figure 1. Due to the “believe-the-positive” rule, the sensitivity of the first follow-up co-testing increased to 92% (95% CI 64–99%) for GynTect^®^/cytology and 100% (95%CI 75–100%) for hrHPV/cytology. In contrast, the specificity decreased to 48% (95%CI 28–68%) for GynTect^®^/cytology and 29% (95%CI 13–50%) for hrHPV/cytology. Considering all follow-up visits, both hrHPV/cytology and GynTect^®^/cytology detected all recurrences (sensitivity 100% (95%CI 75–100%)). The specificity is 43% (95%CI 24–64%) for GynTect^®^/cytology compared to 24% (95%CI 10–46%) for hrHPV/cytology, but this difference is not statistically significant (Figure 1).

## 4. Discussion

The primary aim of this proof-of-concept case-control study was to evaluate the accuracy of GynTect^®^ methylation markers versus hrHPV testing to detect recurrent disease in patients treated for CIN3. Additionally, we explored the accuracy of the clinically relevant co-testing with cytology.

Of the initial 48 patients analyzed, five cases and ten controls were shown to be GynTect-negative at baseline (prior to surgery) and were excluded from accuracy analyses. The rationale for this decision was that GynTect-negative CIN3 would very likely remain GynTect-negative when recurring. These patients would thus not profit from this assay as a test of cure. In the initial cohort, the cumulative probability of recurrence at the end of follow-up was 8.5%. Taking into account the case-control-specific proportions of GynTect-negatives in our present study, we project that about 32% of patients from the target population would be GynTect-negative at baseline. This is in line with the results of previous studies [39,41]. Among young patients aged ≤30, the GynTect-negative proportion is around 47% [41]. In this context, it is also of note that in a recent prospective, multicenter, observational study among patients younger than 30 years, 56% of the GynTect-negative CIN3 regressed within 12 months of watchful waiting, and none of the lesions progressed [42].

For the subgroup of 33 GynTect-positive patients in the current study, the results of the cytology, GynTect^®^, and hrHPV tests at the first follow-up visit and the overall results of all follow-up visits showed that none of the tests on their own could detect all recurrences. There was no significant difference in sensitivity when comparing GynTect^®^ and hrHPV results, irrespective of the number of follow-up visits. However, specificity was significantly higher for GynTect^®^ (90%, 95%CI 71–98%) when compared to hrHPV (62%, 95%CI 40–80%) at the first follow-up visit and was just below significance for all follow-up visits. Reinfection with hrHPV types or persisting multifocal hrHPV infections without clinical disease is most likely responsible for the poorer specificity of the hrHPV test [43,44]. In a co-testing setting, both hrHPV/cytology and GynTect^®^/cytology detected all recurrences. However, the observed higher specificity for GynTect^®^/cytology as compared to hrHPV/cytology is not statistically significant.

A total of 3 of the 12 recurrences were CIN2. Of particular interest is that all three CIN2 were GynTect-negative for all follow-up visits. Moreover, two of the lesions were also hrHPV-negative. Importantly, the HPV results of the current study are based on the detection of 13 hrHPV types only (HPV16, 18, 31, 33, 35, 39, 45, 51, 52, 56, 58, 59, and 68). These HPV types are detected by the most commonly used hrHPV assays [45]. In a post-hoc analysis, detailed HPV genotyping (including HPV26, 53, 66, 67, 70, 73, and 82) of the three CIN2 recurrences revealed HPV53 and HPV67 in two patients during follow-up. For the latter case, HPV16 was also detected during follow-up, but HPV67 was the predominant type. The third lesion remained negative for all tested HPV types. HPV53 and HPV67 are categorized by the IARC as possibly carcinogenic to humans. Since all patients in our study population had surgical excision for HPV16- or HPV18-positive CIN3, both recurrences most likely reflect incident CIN. Moreover, the GynTect-negative result in these two lesions is in line with the low progression potential for HPV53 and HPV67-positive CIN [46,47]. Finally, 9 of the 12 recurrences were CIN3. All of them had the same hrHPV type as the initial lesion indicative of residual disease, and all were detected by the GynTect^®^ assay. In a similar study by Dick and colleagues, 14 of 14 recurrent CIN3 but only 10 of 27 (37%) recurrent CIN2 were detected by the methylation markers FAM19A4/*mi*R124-2 during follow-up [24].

For our proof-of-concept study, we used a case-control design, which is common in the early phase of predictive research. All cases and controls were retrospectively sampled from a single well-characterized prospective cohort study population [9]. Assessment of cytology, hrHPV, and GynTect^®^ (index tests) was blinded to the case-control status. Predictive accuracy was compared in a paired manner where patients underwent all index tests. The small sample size, chosen for feasibility, limits the precision of accuracy estimates and the power of the statistical comparisons. Moreover, 4 out of 21 controls included in the accuracy assessment were missed at the third follow-up visit. For these four patients, the overall test results and the recurrence status were limited to one year of follow-up. Finally, although all controls had a colposcopy, a biopsy was only indicated for five patients. Thus, for controls, the specificity could be biased by misinterpretation of the colposcopic view.

## 5. Conclusions

GynTect^®^ methylation markers and hrHPV were similarly sensitive for the detection of recurrent CIN2/3 lesions during post-treatment surveillance. However, the specificity for the methylation markers was significantly higher. Moreover, the study results show promise that the methylation markers can discriminate between incident CIN and residual disease and thereby may have the potential to spare women from immediate re-conization. A future prospective validation study will have to show whether GynTect^®^/cytology co-testing can outperform hrHPV/cytology co-testing in post-treatment surveillance.

## Figures and Tables

**Figure 1 cancers-16-03022-f001:**
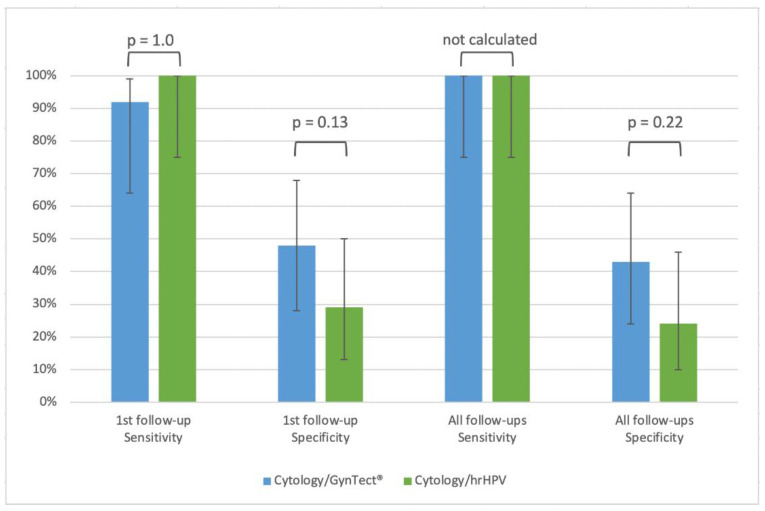
Sensitivity and specificity at first follow-up (single test performance) and over three follow-up visits (program performance) of co-testing for prediction of recurrence during 24 months in 12 cases and 21 controls who were GynTect-positive at baseline.

**Table 1 cancers-16-03022-t001:** Number of patients (*n*) according to test results at first follow-up visit and overall visits in cases and controls who were GynTect-positive at baseline.

Cytology	GynTect^®^	hrHPV	Controls (*n* = 21)		Cases (*n* = 12)	
			1st Follow-Up n	Overall n	1st Follow-Up n	Overall n
Negative	Negative	Negative	6	4	0	0
Negative	Negative	Positive	4	5	1	0
Negative	Positive	Negative	0	1	0	0
Negative	Positive	Positive	2	1	5	5
Positive	Negative	Negative	7	7	2	2
Positive	Negative	Positive	2	2	1	1
Positive	Positive	Negative	0	0	0	0
Positive	Positive	Positive	0	1	3	4

**Table 2 cancers-16-03022-t002:** Sensitivity and specificity at first follow-up (single test performance) and over three follow-up visits (program performance) for prediction of recurrence during 24 months in 12 cases and 21 controls who were GynTect-positive at baseline.

Test-Performance		Cytology	GynTect^®^	hrHPV	*p*-Value GynTect^®^ vs. hrHPV McNemar Exact
Single test 1st follow-up	Sensitivity (95% CI)	50% (25–75%)	67% (39–87%)	83% (55–96%)	0.50
	Specificity (95% CI)	57% (36–76%)	90% (71–98%)	62% (40–80%)	0.03
Overall 1st to 3rd follow-up	Sensitivity (95% CI)	58% (31–81%)	75% (46–92%)	83% (55–96%)	1.00
	Specificity (95% CI)	52% (32–72%)	86% (65–96%)	57% (36–76%)	0.07

## Data Availability

All data generated in this study are available from the corresponding author upon reasonable request. Privacy and ethical restrictions will be accounted for.

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
