# Peer review of "Accuracy of GynTect® Methylation Markers to Detect Recurrent Disease in Patients Treated for CIN3: A Proof-of-Concept Case-Control Study"

_cancers, 2024, doi:10.3390/cancers16173022_

Round 1

Reviewer 1 Report

Comments and Suggestions for Authors

In this study the authors describe the accuracy of methylation markers to detect recurrent CIN III compared with the detection-accuracy of HPV/cytology testing alone. Sensitivity was similar for both methods, while methylation testing showed a better specificity, resulting in less false positives.

This is a well written manuscript with a clear message. My main criticism/advise is that the authors should try to better explain what these results add to the current literature. There are already several published studies showing the same results: what exactly is the difference between the results from the current study compared with published literature? 

Author Response

This is a well written manuscript with a clear message.

Query: My main criticism/advise is that the authors should try to better explain what these results add to the current literature. There are already several published studies showing the same results: what exactly is the difference between the results from the current study compared with published literature?

Reply: In most countries primary cervical cancer screening is based on high-risk HPV detection followed by a triage of HPV positive cases with cytology. Methylation markers show great promise for a more efficient triage approach. Indeed, most publications focus on the use of methylation markers as triage tools. However, to our knowledge there is only one published paper which evaluates the use of methylation markers for the detection of recurrent disease after surgical intervention in a longitudinal setting. This cited study is by Dick and colleagues and uses a different set of methylation markers. We refer to this study in the introduction as well as in the discussion.

Reviewer 2 Report

Comments and Suggestions for Authors

The authors presented the study entitled” Accuracy of GynTect® Methylation Markers to detect recurrent disease in patients treated for CIN3: A proof-of-concept case control study” by including limited cases number, controls 31 and cases 17.

I have suggestions. CIN cases related risk factors should be listed by tables. No appropriate comparisons result for characteristics of patients between the control and cases group. Control group is based on what method of a random feasibility sample of controls.

Author Response

Query: CIN cases related risk factors should be listed by tables.

Reply: We have added endocervical crypt involvement as another risk factor for recurrence in the introductory section of the manuscript. We prefer not to include a table in which all risk factors are listed. We feel that this would distract the reader from the main aim of the study.

Query: No appropriate comparisons result for characteristics of patients between the control and cases group.

Reply: The table listing the characteristics of the patients of both groups is a supplementary table (S1). It was only included in the main text of the manuscript to ease the review process. In the final version it will have to be downloaded as a separate file. We decided to include this table to demonstrate the homogeneity of both groups with respect to age, number of births and other parameters. The only notable difference concerns the state of the excision margins which is not unexpected. The margins were more frequently positive among cases compared to controls. This difference, however, is not significant (p=0.23, Fisher´s exact test). This information is now included in materials and methods.

Query: Control group is based on what method of a random feasibility sample of controls.

Reply: Ten percent of the original cohort without recurrence were used as controls. The ratio of controls to cases was thus about 2 to 1. In practise, every 10th patient without recurrence was chosen in descending order from the list of the original cohort. The sentence in the manuscript is rephrased accordingly.

Reviewer 3 Report

Comments and Suggestions for Authors

The authors has performed a study regarding the analysis of DNA methylation (GynTect) compared to regular screening by hrHPV/cytology co-testing to evaluate the performance of the methylation test for the detection of recurrent CIN2/3 42 during follow-up. This might be an interesting study. Though, there are several issue that need to be addressed.  

Major remarks:

-          In the introduction they describe a lot about the viral DNA integration study. This can be shortened.

-          The authors do describe that the study is very small, but overfitting is really to be considered, as they only have 12 cases left. The authors should consider whether additional samples from other study cohorts should be included.

-          In the results they also should look at HPV16 and/or HPV18 as a test. They do describe the specific HPV genotypes in the discussion, but this should be already presented in the results.

-          What is the (mean) duration that the patients will get a recurrence? This should be added. Might also consider to show methylation, HPV positivity in relation to the duration?

-          Also show the sensitivity / specificity of non-free margins.

-          In the overall numbers how many women get the 12 and/or 24 months test? Please adjust the table into 6, 12 and 24 months.

-          Please rephrase sentence (line 259-260) “Specificity for GynTect®/cytology was somewhat better than for hrHPV/cytology but this difference was not statistically significant.” If not significantly different than also not somewhat better.

Author Response

Query: In the introduction they describe a lot about the viral DNA integration study. This can be shortened.

Reply: Viral DNA integration is a characteristic step in cervical carcinogenesis. In a previous study we evaluated the potential of HPV DNA integration as an individualized molecular marker for the detection of recurrent CIN after surgery. Since this cohort served as the basis for the current study, we feel that it is necessary to inform the reader in brief about the aim and outcome of that study. Shortening of this passage (7 lines) may well reduce comprehension.

Query: The authors do describe that the study is very small, but overfitting is really to be considered, as they only have 12 cases left. The authors should consider whether additional samples from other study cohorts should be included.

Reply: We agree, that the number of cases is low. Unfortunately, we do not have access to another cohort of patients with a similar follow-up setting. Indeed, the fact that all cases and controls of the present study were recruited for a single well characterized prospective cohort study population may also be interpreted as one of the strengths of the study.

Query: In the results they also should look at HPV16 and/or HPV18 as a test. They do describe the specific HPV genotypes in the discussion, but this should be already presented in the results.

Reply: The results focus on the hrHPV group as such (13 hrHPV types). The reason for this is that many clinics use hrHPV group specific assays for follow-up care of patients after CIN3 surgery. Moreover, most of the literature evaluating the use of HPV-detection and cytology either alone or in a co-testing format are not based on HPV genotyping. In the present study, HPV genotyping was only used for a post-hoc analyses of the cases, the outcome of which is described in the discussion section. To make this strategy clearer, we have added more detailed information in the materials and method section.

Query: What is the (mean) duration that the patients will get a recurrence? This should be added. Might also consider to show methylation, HPV positivity in relation to the duration?

Reply: The median duration for the diagnosis of a recurrence was 6 months (range 3 to 24 months). We have modified materials and methods accordingly. The dynamics of methylation and HPV-positivity over time can be deduced from Table1.

Query: Also show the sensitivity / specificity of non-free margins.

Reply: The sensitivity and specificity of non-free margins for recurrent disease (as listed in Supplementary Table 1) is 29% and 88%, respectively.  Moreover, non-free margins among cases were more frequent than among controls (29% vs 12%) but this difference is not statistically significant (p=0.23, Fisher´s exact test). This data is now included in the materials and methods section.

Query: In the overall numbers how many women get the 12 and/or 24 months test? Please adjust the table into 6, 12 and 24 months.

Reply: We prefer to leave Table 2 as it is in order to avoid confusion. The table has a focus on the test outcome after 6 months and the overall program performance. It should also be considered that most of the controls (17/21) had a 24 months test.  Moreover, the dynamics of the methylation status and HPV positivity over time are shown in Table 1.

Query: Please rephrase sentence (line 259-260) “Specificity for GynTect®/cytology was somewhat better than for hrHPV/cytology but this difference was not statistically significant.” If not significantly different than also not somewhat better.

Reply: The sentence is rephrased to: “However, the observed higher specificity for GynTect®/cytology as compared to hrHPV/cytology is not statistically significant„

Round 2

Reviewer 2 Report

Comments and Suggestions for Authors

Congratulations! Your revised article reaches the requirement for publication.